Label-free quantitative proteomic analysis reveals potential biomarkers for early healing in cutaneous leishmaniasis

Montoya Andrés 1
López Manuel Carlos 2
http://orcid.org/0000-0003-4745-7385 Vélez Ivan D. 1
http://orcid.org/0000-0003-2752-4931 Robledo Sara M. 1 sara.robledo@udea.edu.co
1 PECET, Facultad de Medicina, Universidad de Antioquia , Medellin, Antioquia , Colombia
2 Molecular Biology Department Consejo Superior de Investigaciones Científicas, Instituto de Parasitología y Biomedicina “López Neyra” , Granade , Spain
Braga Erika
Electronic publication date: 2019 Jan 11
Publication date: 2019
Volume: 6
Electronic Location ID: e6228
Received 2018 Sep 27; Accepted 2018 Dec 6
Copyright: © 2019 Montoya et al.
Copyright year: 2019
Copyright holder: Montoya et al.
License: This is an open access article distributed under the terms of the Creative Commons Attribution License, which permits unrestricted use, distribution, reproduction and adaptation in any medium and for any purpose provided that it is properly attributed. For attribution, the original author(s), title, publication source (PeerJ) and either DOI or URL of the article must be cited.
License URL: https://creativecommons.org/licenses/by/4.0/

Keywords: Therapeutic response, Biomarkers, Cutaneous leishmaniasis, Label-free proteome

Funding: El Departamento Administrativo de Ciencia, Tecnología e Innovación–Colciencias CT-695-2014 This work and Andrés Montoya received financial support from El Departamento Administrativo de Ciencia, Tecnología e Innovación–Colciencias (CT-695-2014). The funders had no role in study design, data collection and analysis, decision to publish, or preparation of the manuscript.

==============================
Background

Leishmaniasis is a parasitic disease caused by more than 20 species of the Leishmania genus. The disease is globally distributed and is endemic in 97 countries and three territories in the tropical and subtropical regions. The efficacy of the current treatments is becoming increasingly low either due to incomplete treatment or resistant parasites. Failure of treatment is frequent, and therefore, the search for early biomarkers of therapeutic response in cutaneous leishmaniasis (CL) is urgently needed.

Objective

The aim of this study was to compare the proteomic profiles in patients with CL before and after 7 days of treatment and identify early biomarkers of curative response.

Methods

Four patients with a parasitological diagnosis of leishmaniasis with confirmation of species by PCR-RFLP were recruited. All patients had a single lesion, and a protein from the middle of the ulcer was quantified by liquid chromatography and mass spectrometry.

Results

A total of 12 proteins showed differential expression in the comparative LC-electrospray ionization MS/MS (LC-ESI-MS/MS) triplicate analysis. Seven of them were up-regulated and five of them were down-regulated. Calcium binding proteins A2, A8, and A9 and hemoglobin subunits alpha-2 and delta showed high correlation with epidermis development and immune response.

Conclusion

We identified changes in the profiles of proteins that had a positive therapeutic response to the treatment. The proteins identified with differential expression are related to the reduction of inflammation and increased tissue repair. These proteins can be useful as biomarkers for early monitoring of therapeutic response in CL.

Introduction

Leishmaniasis is a parasitic disease caused by more than 20 species of Leishmania genus. The parasite is transmitted by Diptera insects of the genus Phlebotomus (Old World) and Lutzomyia (New World) (Bates, 2007). The disease is distributed on five continents and is endemic to 97 countries and three territories in the tropical and subtropical regions (Alvar et al., 2012). Cutaneous leishmaniasis (CL), which is the most prevalent clinical form, manifests as a single or multiple ulcer with indurated edges, is painless, and presents a granular background that may or may not be covered by an exudate; the presence of plaques and nodules is also frequent (Handler et al., 2015).

Despite the high number of people affected, leishmaniasis control programs present difficulties not only with the diagnosis of cases and the control of vectors to reduce transmission but also with the treatment, given that the therapeutic response with the available drugs is becoming poorer due to incomplete treatments or the development of resistance in the parasites (Oliaee et al., 2018). Unfortunately, the confirmation of a definitive cure is complicated by the possibility of relapses after an initial cure, making necessary long follow-up periods (up to 6 or 12 months).

The prognosis of the treatment can be improved by detecting an early curative response or treatment failure. Such markers should also specifically correlate with the response to treatment and predict the long-term clinical outcome, by non-invasive sampling methods.

Up to now, the main advances have been made in visceral leishmaniasis (VL) where levels of IgG1 and IgG2 antibodies, TNFα, IL10, and parasite load were proposed as biomarkers of infection (Portela et al., 2018; Martinez-Subiela et al., 2017; Corpas-lópez et al., 2016). IFN-γ, IL-2, monokine-induced-by-IFN-γ (MIG) and IFN-γ-inducible protein-10 (IP-10) could be used as markers of cure in immunocompetent patients with VL caused by Leishmania infantum (Ibarra-Meneses et al., 2017). In case of CL, the levels of IL4, IL8, INFγ, and MMP2 in skin biopsies and levels of TNFα, IL10, IL15, IL32γ, TGFβ, sCD26, sCD30, Cortisol, Prolactin, and SOD1 in serum, have been associated with infection and disease susceptibility (Kammoun-rebai et al., 2016; Kip et al., 2015; Carvalho et al., 2017). Nevertheless, these candidates are not fully validated as biomarkers of infection, and other biomarkers for treatment monitoring have not been identified or validated.

Analysis by liquid chromatography and electrospray ionization in tandem mass spectrometry (LC-ESI-MS/MS) has emerged as one of the most robust alternatives for the comparison of proteomic profiles through a quantitative approach based on labels (Ong & Mann, 2005). Quantitative analysis has been useful in clinical proteomics, proteomics of subcellular protein, and high-resolution comparative proteomics (Bantscheff & Lemeer, 2012), which has advanced the analysis of systems biology, particularly in the elucidation of the mechanisms that underlie the pathophysiology of several diseases.

The introduction of stable isotopes is the basis for the quantification in label-based approach and various protocols are now available that vary depending on the used methods for the incorporation of isotopes in the peptides/proteins. One of these methods is the isobaric tag for relative and absolute quantification (iTRAQ). The iTRAQ is a label-based method that allow the simultaneous analysis of eight samples in a single run of MS (multiplexing), resulting in reduced analytical variability (Pottiez et al., 2012). With this method, the labeled peptides are shown as a single peak in the MS spectrum and, after fragmentation of peptides in the MS/MS analysis, the isotopes are released and peptides are identified according to their masses. This type of proteomic analysis results in data with high quality, sensitivity, and a wide dynamic range (Thompson et al., 2003).

The use of unlabeled protein profiles to examine differentially expressed proteins in tissue samples largely eliminates the variations and biases in replicate MS measurements and has allowed for the exploration of the mechanism of tumorigenesis and the discovery of biomarkers in various diseases (Griffin et al., 2010; Atrih et al., 2014).

Based on the advantages offered by this methodology, the aim of this study was to find and validate early biomarkers of therapeutic response in CL using proteomics without labeling in order to provide patients with adequate monitoring and identify early treatment failure, thus providing the possibility of changing the therapeutic scheme in a timely manner.

Materials and Methods

Clinical samples

In this study we included samples from patients with parasitological diagnosis of CL caused by Leishmania panamensis as confirmed by PCR-RFLP, with a single lesion, an evolution period of less than 4 months, and a lesion area less than four cm2. The proteomic analysis was done with samples from four patients while the RT-qPCR analysis was carried out in samples from five patients.

Sample preparation for proteomic analysis

Granulation tissue from the center of the ulcer was collected before starting the treatment (TD0) and on day 7 of treatment (TD7). The material was placed in a vial containing 50 mM Tris, 100 mM NaCl, 20 mM DTT, protease inhibitor, and 1× phosphatases, with pH 7.5 buffer. The samples were sonicated for 5 min at 4 °C and subsequently centrifuged at 13,000 rpm for 20 min at 4 °C; the supernatants were stored at −80 °C until use. The proteins were quantified, and their integrity was confirmed by SDS-PAGE.

Immunoaffinity depletion

Protein extracts were diluted 5-fold with 1× dilution buffer (Tris-Buffered Saline—10 mM Tris–HCl with 150 mM NaCl, pH 7.4), filtered using a 0.45 μm pore-size spin filter to remove particulate materials and centrifuged at 9,000 × g for 1 min. Each of the diluted and filtered samples (500 μl) were injected in a Seppro IgY 14 LC5 column with a constant flow rate of 0.5 ml/min for 20 min, followed by a Seppro washing step at a flow rate of 2.0 ml/min for 3 min. Non-bound proteins (depleted fraction) were collected in the flow-through fraction. Due to the large volume of collected fractions, depleted samples were concentrated using Amicon™ Ultra-15 Centrifugal Filter Units (Millipore, Burlington, MA, USA). Concentrated samples were reconstituted in a chaotropic buffer containing 8M urea, 2M thiourea, and 100 mM triethylammonium bicarbonate at pH 8.5. Concentrated fractions were stored at −80 °C (Hyung et al., 2014).

In-gel protein digestion (Stacking gel)

An aliquot concentrated flow-through fraction of each sample from four patients was diluted with enough amount of sample loading buffer and then applied onto 1.2 cm wells of a conventional SDS-PAGE gel (1 mm thick, 4% stacking, and 12% resolving). The run was stopped when the sample front entered one cm into the resolving gel, to concentrate the whole proteome in the stacking/resolving gel interface. The bands of unseparated protein were visualized by Coomassie staining and then, bands were excised (Andreu et al., 2017), cut into pieces (1 mm2), deposited in 96-well plates, and processed automatically in a Proteineer DP (Bruker Daltonics, Bremen, Germany). For digestion step, gel pieces were washed with 50 mM ammonium bicarbonate (NH4HCO3) and then, with ACN prior to reduction with 10 mM DTT in 25 mM NH4HCO3. The alkylation was carried out with 55 mM IAA in 50 mM NH4HCO3. Gel pieces were then rinsed firstly with 50 mM NH4HCO3 and secondly with ACN and then were dried under a stream of nitrogen. Proteomics grade Trypsin (Sigma Aldrich) at a final concentration of 16 ng/μl in 25% ACN/50 mM NH4HCO3 was added and the digestion took place at 37 °C for 4 h. The reaction was stopped by adding 50% ACN/0.5% TFA for peptide extraction. The tryptic eluted peptides were dried by speed-vacuum centrifugation and were cleaned up/desalted using Stage-Tips with Empore 3M C18 disks (Sigma-Aldrich, St. Louis, MO, USA).

Liquid chromatography and mass spectrometer analysis

One μg aliquot of the previously digested sample from four patients was subjected to 1D-nano LC ESI-MS/MS analysis using a nano-liquid chromatography system (Eksigent Technologies nanoLC Ultra 1D plus; SCIEX, Foster City, CA, USA) coupled to high-speed Triple TOF 5600 mass spectrometer (SCIEX, Foster City, CA, USA) with a Nanospray III source. A silica-based reversed phase Acquity UPLC M-Class Peptide BEH C18 Column, 75 μm × 150 mm, 1.7 μm particle size, and 130 Å pore size (Waters, Milford, MA, USA), was used as analytical column. The trapping column was a C18 Acclaim PepMapTM 100 (Thermo Scientific, Waltham, MA, USA), 100 μm × 2 cm, five μm particle diameter, and 100 Å pore size. The loading pump delivered a solution of 0.1% formic acid in water at two μl/min. The nano-pump provided a flow-rate of 250 nl/min and was operated under gradient elution conditions. Peptides were separated using a 250 min gradient ranging from 2% to 90% mobile phase B (mobile phase A: 2% acetonitrile, 0.1% formic acid; mobile phase B: 100% acetonitrile, 0.1% formic acid). Injection volume was five μl.

A TripleTOF 5600 System (SCIEX, Foster City, CA, USA) was used for data acquisition with an ion spray voltage floating 2300 V, curtain gas 35, interface heater temperature 150, ion source gas 1 25, declustering potential 100 V. All data were acquired using information-dependent acquisition (IDA) mode with Analyst TF 1.7 software (SCIEX, Foster City, CA, USA). For IDA parameters, 0.25 s MS survey scan in the mass range of 350–1250 Da were followed by 35 MS/MS scans of 100 ms in the mass range of 100–1,800 (total cycle time: 4 s). Switching criteria were set to ions greater than mass to charge ratio (m/z) 350 and smaller than m/z 1,250 with the charge state of 2–5 and an abundance threshold of more than 90 counts (cps). Former target ions were excluded for 15 s. IDA rolling collision energy (CE) parameters script was used for automatically controlling the CE (Segura et al., 2013).

Quantification and data analysis

The mass spectrometry data obtained were processed using PeakView® 2.2 Software (SCIEX, Foster City, CA, USA) and exported as mgf files. Proteomics data analysis were performed by using four search engines (Mascot Server v.2.6.1, OMSSA, X!Tandem, and MyriMatch) and a target/decoy database built from sequences in the Homo sapiens reference proteome at UniProt Knowledgebase. Search parameters included: trypsin as enzyme; two allowed missed cleavages; carbamidomethyl (C) as a fixed modification and acetyl (Protein N-term) and Oxidation (M) as variable modifications; peptide mass tolerance ± 25 ppm for precursors and 0.02 Da for fragment masses. Score distribution models were used to compute peptide-spectrum match p-values, and spectra recovered by a false discovery rate (FDR) <= 0.01 (peptide-level) filter were selected for quantitative analysis (Li et al., 2017a).

Label-free quantification

Differential regulation was measured using linear models and the q-values (FDR) were used for determining statistical significance. All analyses were conducted using Proteobotics software (Madrid, Spain) (Griffin et al., 2010).

Pathway and interaction network analysis

The identified proteins were analyzed for biological function through the Panther Classification System website (http://www.pantherdb.org). The proteins were categorized according to their components, functions, and biological processes. The protein–protein interaction network was constructed using the online STRING database (https://string-db.org) (Wu, Hasan & Chen, 2014).

RNA isolation

Five patients were included in the validation assay. The RNA was extracted using High Pure RNA Tissue Kit (Roche, Mannheim, Germany), from scraping samples obtained on TD0 and TD7. Samples were placed in 400 μl of lysis/binding buffer and sonicated for 5 min in cold room. Then, 200 μl ethanol was added and each sample was transferred to a high pure filter tube. After 30 min centrifugation at 13,000 × g, samples were treated with DNase for 15 min at 25 °C. A total of 500 μl wash buffer I were added and samples were centrifuged during 30 s at 8,000×g. Washing step was repeated using 300 μl of wash buffer II and centrifugation for 30 s at 13,000×g. Finally, the RNA was eluted with 30 μl of elution buffer. A reverse transcription reaction was performed in a 20 μl volume in a PTC-100 Programmable Thermal Controller (MJ Research, Inc., Quebec, Canada) using the Transcriptor First Strand cDNA Synthesis kit (#04379012001; Roche Diagnostics, Risch-Rotkreuz, Switzerland) according to the manufacturer’s instructions, with certain modifications.

cDNA synthesis

A reverse transcription reaction was performed in 20 ng of RNA using the maximum first strand cDNA synthesis kit, following the manufacturer’s instructions: four μl master mix reaction, two μl RT enzyme mix, two μl RNA (DNase I treated), and 12 μl water were mixed and incubated in the PTC 100TM (MJ research Inc., St. Bruno, Quebec, Canada) thermocycler for three cycles: 10 min at 25 °C, 15 min at 50 °C, and 5 min at 85 °C. The obtained cDNA (20 μl) was stored at −20 °C.

RT-qPCR

Each reaction mixture consisted of two μl reverse transcription reaction product as the template, 0.5 μl each of forward and reverse primers, seven μl PCR grade water and 10 μl of QuantiTect SYBR Green PCR Kits (Qiagen, Hilden, Germany). The total volume of the reaction was 20 μl. The RT-qPCR reaction was completed with an initial denaturation step (95 °C for 15 min) followed by 40 cycles of denaturation (95 °C, 10 s), annealing (60 °C, 30 s) and extension (72 °C, 30 s). A final cooling step was conducted at 72 °C for 300 s. The RT-qPCR reaction was performed using the LightCycler 96 Instrument. Each sample was analyzed in duplicate.

Three genes from up regulated proteins SERPINB1, S100A8, and S100A9 were selected for the measurement of relative expression. The glyceraldehyde-3-phosphate dehydrogenase (GAPDH) gene was used as constitutive gene. The primers were designed using Primer 3 software (bioinfo.ut.ee/primer3-0.4.0/). The following primer sequences were used: SERPINB1, Fwd 5′-ATGAAAGAAGCCACGACGAA-3′ and Rev 5′-GGTAAGGCAGTTCCAGCACA-3′; S100A8, Fwd 5′-GCCAAGCCTAACCGCTAT-3′ and Rev 5′-CCCACCTGAAAAACAGAACC-3′; S100A9, Fwd 5′-GCACCCCCTAAGACCACAG-3′ and Rev 5′-CCCCACAGCCAAGACAGTT-3′; and GAPDH, Fwd 5′-GCTTCGCTCTCTGCTCCTC-3′ and Rev 5′-ACGACCAAATCCGTTGACTC-3′.

Ethics statement

All study subjects provided their informed written consent to be included in the study. The study was approved by the Human Bioethics Committee of Universidad de Antioquia–SIU (Act 007-2017).

Results

All patients were treated with meglumine antimoniate. The cure was monitored for 6 months (until PTD 180). By this time all patients healed their lesion. No reactivation was detected (Fig. 1).

Figure 1 Evolution of ulcer in patients.

The figure shows the appearance of the cutaneous leishmanaisis during follow-up. TD0 (A, D, G, J): before the start of treatment; TD7 (B, E, H, K): on day 7 of treatment; PTD180 (C, F, I, L): 6 months after the end of treatment.

Proteins detected in granulation tissue of CL

In total, 344 proteins were identified in the granulation tissue formed in the center of the ulcer. After analyzing the quality of the data obtained in MS, three of the four biological replicas were selected for label-free quantification by Q-values (FDR) using Proteobotics software (Madrid, Spain). The label-free quantification analyses yielded 267 proteins that were truly quantified in the three biological samples (p < 0.05). Of the 267 proteins, 86 proteins were up-regulated (32.21% ratio > 0.67), 59 proteins were down-regulated (22.09% ratio <0.717) and 122 proteins showed no changes (44.9%). The 267 proteins were grouped according to their cellular locations as follows: 63 intracellular, four intracellular membranes, 53 extracellular membranes, six extracellular matrices, 56 extracellular space, 43 macromolecular complex and 42 organelles (Fig. 2A). The proteins were also classified according to their Class: 21 Calcium binding protein, 11 Chaperone, 13 Cytoskeletal protein, 14 Defense/immunity protein, 33 Enzyme modulator, 18 Hydrolase, 10 Membrane traffic protein, 13 Nucleic acid binding, 22 Receiver, and 112 Other (Fig. 2B). Finally, they were also classified according to their biological process: Antioxidant activity 7, Binding protein 89, Catalytic activity 59, Receptor activity 2, Structural molecule activity 13, Translation regulatory activity 1, Transporter activity 5, and Other 91 (Fig. 2C).

Figure 2 Global proteomic analysis of the granulation tissue.

Samples were obtained from a cutaneous leishmaniasis ulcer before treatment (TD0) and day 7 of treatment (TD7). The figure shows the gene ontology analysis of all the quantified proteins. (A) Proteins present in cellular components. (B) Protein classified in classes. (C) Molecular function of proteins.

Label-free quantification

Using the FDR analysis, 12 proteins were found with differential expression when comparing TD0 and TD7. Five were down-regulated (FABP5, S100A2, PRDX2, HBA1, and HBD) and seven were up-regulated (KRT73, LAMP2, S100A9, S100A8, KRT10, SERPINB1, and ATP7A). All differentially expressed proteins were classified using the Panther Classification System website (http://www.pantherdb.org).

The classification according to their molecular function showed a distribution between fatty acid binding 3, Toll-like receptor 4 binding 2, arachidonic acid binding 2, RAGE receptor binding 2, and oxygen transporter activity 2 (Fig. 3). The classification of the cellular components showed that all proteins are located at the extracellular level: extracellular region 10 genes, extracellular exosome nine genes, extracellular region 10 genes, extracellular space six genes and membrane-bound vesicle eight genes.

Figure 3 Molecular function of differentially expressed proteins early.

The figure shows the classification of the 12 proteins with differential expression according to their molecular function. Pathway description.

The classification according to their biological process showed that these proteins participate in neutrophil aggregation and degranulation, sequestering of zinc, peptidyl-cysteine S-nitrosylation, chemokine production, leukocyte migration during the inflammatory response, removal of superoxide radicals, oxygen transport, hydrogen peroxide catabolic process, astrocyte development, antimicrobial humoral response, and epidermis development. The proteins showed a fold enrichment between 17.1% and 100%, the p-log10 value between 3.9 and 5.7 and an FDR <0.05 (Table 1).

Table 1 Classification of proteins with differential expression according to the complete biological process*.

GO biological process complete	Genes	−Log10 (p-value)	Fold enrichment	FDR	
Neutrophil aggregation (GO:0070488)	P06702,P05109	5.7	>100	0.00464	
Sequestering of zinc ion (GO:0032119)	P06702,P05109	5.4	>100	0.00696	
Peptidyl-cysteine S-nitrosylation (GO:0018119)	P06702,P05109	5.1	>100	0.00683	
Chemokine production (GO:0032602)	P06702,P05109	4.7	>100	0.0122	
Leukocyte migration involved in inflammatory response (GO:0002523)	P06702,P05109	4.6	>100	0.0134	
Removal of superoxide radicals (GO:0019430)	P32119,Q04656	4.5	>100	0.0152	
Oxygen transport (GO:0015671)	P69905,P02042	4.4	>100	0.0163	
Hydrogen peroxide catabolic process (GO:0042744)	P32119,P69905	4.2	>100	0.0242	
Astrocyte development (GO:0014002)	P06702,P05109	3.9	>100	0.0426	
Antimicrobial humoral response (GO:0019730)	P06702,P05109,Q04656	4.6	52.08	0.0134	
Neutrophil degranulation (GO:0043312)	Q01469,P13473,P06702,P05109,P30740	5.3	18.15	0.00643	
Epidermis development (GO:0008544)	Q04656,P13645,Q01469,Q86Y46	4.2	17.11	0.0233	
Note:

* Panther classification system.

Interaction network analysis

The proteins were analyzed to determine the functional interaction network that unites the 12 proteins differentially expressed between TD0 and TD7 using STRING database, obtaining a minimum required interaction score of 0.4 (medium confidence). The analysis showed that 5 of the 12 proteins had a good correlation and were classified in two groups. In the first group were S100A9, S100A8, and S100A2, all calcium-binding proteins. In the second group were HBD and HBA2, both components of hemoglobin (alpha and delta) (Fig. 4).

Figure 4 Interaction network analysis of protein.

Functional interaction network linking the significantly differentially expressed proteins between TD0 and TD7. S100A9, S100 calcium binding protein A9; S100A8, S100 calcium binding protein A8 up-regulated; S100A2, S100 calcium binding protein A2; HBD, Hemoglobin, delta; HBA2, Hemoglobin, alpha 2 down-regulated.

Relative expression by RT-qPCR

The levels of expression of SERPINB1, S100A8, S100A9 at TD7 were compared with respect to the levels before treatment (TD0) using the ΔΔCT method. All five patients were cured after 6 months of follow-up. The expression levels increased 2.65 ± 1.22 times for SERPINB1, 3.2 ± 2.81 for S100A8, 1.38 ± 0.67 for S100A9 (Fig. 5).

Figure 5 Fold induction of S100A8, S100A9 and SERPINB1 by RT-qPCR.

The figure shows the induction of each gene at day 7 of treatment vs. before treatment calculated by the ΔΔCT method. S100A8 (circles). S100A9 (squares). SERPINB1 (triangle). Each point corresponds to one patient.

Discussion

Leishmaniasis is one of the seven main tropical diseases and continues to be one of the major public health problems around the world (Torres-Guerrero et al., 2017). Re-evaluation of treatment guidelines for CL is urgently needed because there is high risk of toxicity of pentavalent antimony and treatment failure is increasing (Von Stebut, 2015; López-Carvajal, Mazo & Cardona-Arias, 2016).

The major adverse effects associated with pentavalent antimonial treatments are local pain, arthralgia, myalgia, increases in hepatic enzymes, urea, and creatinine, and electrocardiographic alterations (inversion of the T wave, prolongation of the Q-T segment, depression of the S-T segment, and sinus bradycardia). Based on their severity, it is necessary to stop the treatment and modify the treatment scheme or change the drug (Romero et al., 2017).

The search for biomarkers in leishmaniasis has emerged as a need to not only identify markers of susceptibility and resistance to infection or the evaluation of vaccines effectiveness, but also monitor early and late therapeutic efficacy (Portela et al., 2018; Martinez-Subiela et al., 2017; Corpas-lópez et al., 2016; Kammoun-rebai et al., 2016). However, none of the proposed biomarkers has been validated.

For monitoring of the therapeutic response of CL, various biomarkers were proposed. These biomarkers include the parasite load, levels of IL4, IL8, INFγ, MMP2 expression in skin biopsies and also levels of TNFα, IL10, IL15, IL32γ, TGFβ, sCD26, sCD30, cortisol, prolactin, and SOD1 in serum (Kip et al., 2015). Nevertheless, the applicability of these proposed biomarkers is still debatable either because of the difficulty of collecting samples from invasive biopsies, or because the cytokines detected in serum cannot be attributed to a specific infectious process located on the skin as would be in the case of CL, where it can be influenced by conditions particular to each individual.

Little has been explored in the search for early biomarkers of therapeutic response to predict the effectiveness of treatment in the first week. The biomarkers mentioned above were detected in skin biopsies and serum samples before treatment and their previous levels correlated with the therapeutic response at the end of treatment. In our approach, we use a sample obtained of the center of the ulcer by scraping that does not generate pain or discomfort in the patient. Additionally, the material is obtained from the site where both the damage and repair process occur, during a curative therapeutic response.

The healing proteome has been studied in different animal models, especially in mice and pigs. These studies showed that proteins with differential expression correspond to receptors, hydrolases, enzymes, immunity, cytoskeletal proteins, transporters, and oxidoreductases (Chen et al., 2017; Sabino et al., 2015). Additionally, in pathogenic processes such as type 2 diabetes and chronic ulcers, where there is an increase in expression levels of proteins related to inflammation and tissue destruction, proteins with differential expression have been identified, most of which relate to angiogenesis, cell death, and inflammation (Eming et al., 2010; Krisp et al., 2013).

In CL, where scarring of the lesion is distorted by the presence of the parasite and the inflammatory phase is perpetuated over time, the healing process only begins with the disappearance of the antigenic stimulus triggered by the parasite.

Our results showed differential expression in 12 proteins when comparing expression levels on TD0 and TD7. Five were down-regulated and seven were up-regulated. The up-regulated proteins participate in biological processes such as neutrophil aggregation, response to zinc ion, sequestering of zinc ion, peptidyl-cysteine S-nitrosylation, and cellular transition metal ion homeostasis, the down-regulated proteins participate in oxygen transport and hydrogen peroxide catabolism.

There was a decrease in the expression of FABP5 and S100A2. FABP5 is a fatty acid binding protein while S100A2 is a calcium transport protein. Both proteins show increased levels in psoriasis (Chamcheu et al., 2014), revealing their role in the inflammatory process (Eckert et al., 2004). Thus, the decrease in FABP5 and S100A2 in the case of CL would correlate with a regulation of the inflammatory stage. On the other hand, PRDX2 is a peroxidase which has been considered a biomarker in inflammatory processes such as multiple sclerosis (Voigt et al., 2017). Its early decline in CL also confirms a regulation of the inflammatory process. Finally, HBA1 and HBD are subunits of hemoglobin, which is highly important in oxygen transport that favors oxidative response. Decrease of HBA1 and HBD levels in tissue can be correlated with a reduction of the oxidative response and therefore of the damage generated in the tissue by oxidative mechanisms. Else, the low levels of these proteins in skin could be related to their inability to regulate oxidative response similar to that observed in steatohepatitis (Liu et al., 2011).

Change in the protein profiles with increased keratin, type II cytoskeletal 73, a protein present in the hair follicle (related to keratinization and important in the development of the skin in mice (Liu et al., 2015), confirms that this protein would be participating early in the new tissue formation. On the other hand, LAMP2 is a glycoprotein that has been described as important in processes mediated by lysosomes. In the context of healing, it has been described in squamous cell carcinoma where its expression is related to cell adhesion and cell migration, generating a layer of glycocalyx that prevents the degradation of cells and the action of different proteases (Li et al., 2017b). Thus, in the case of CL it may be participating in the process of tissue repair, possibly in cell migration.

S100A9 and S100A8 are calcium- and zinc-binding proteins related to different processes such as the regulation of inflammation, cell cycle progression and cell differentiation, induction of neutrophil chemotaxis, phagocytosis, as well as degranulation (Zhong et al., 2016). Additionally, they are overexpressed during the healing process (Eming et al., 2010), particularly in the differentiated keratinocytes and their participation in the reorganization of the keratin cytoskeleton in the injured epidermis is well-established (Thorey et al., 2001). Based on their roles in healing, we believe that their increase in early response to treatment correlates with the beginning of the process of tissue repair.

KRT10 is a type of keratin that has been described as an early marker of daughter cell differentiation in the basal stratum that faces the epidermal surface and contributes to re-epithelialization. Studies in mice confirmed that the absence of cells expressing KRT10 correlates with deterioration in re-epithelialization (Zhao et al., 2016). Therefore, in CL, the increase in the expression of KRT10 would be correlated with the beginning of the re-epithelialization process. SERPINB1 belongs to the superfamily of serine protease inhibitors, a protein that has various functions. In the inflammatory process it has been shown to play an important role inhibiting apoptosis, activation of caspase 3 and the action of different proteases such as elastase, cathepsin G and D, and proteinase-3 (Loison et al., 2014). It has been demonstrated in models of corneal epithelium in cattle that during cicatrization SERPINB1 participates in cell migration and its increased level is related to the closure of the wound; the levels are restored to normal values when the injury has healed (Torriglia, Martin & Jaadane, 2017). Both mechanisms would be important in the therapeutic response in CL since increased apoptosis has been described as one of the pathogenesis markers in proteome studies of pathogenesis in CL (Da et al., 2015). On the other hand, cell migration and wound closure would be related to the start of the lesion healing.

Finally, ATP7A is a very interesting protein for the regulation of intracellular copper concentrations, fulfilling two very important functions: the elimination of the accumulated intracellular copper and the transportation of the copper necessary for the proper functioning of the copper-dependent enzymes (Da et al., 2015). The human peptide Gly- (L-His)–(L-Lys) or GHK has a high affinity for copper as a carrier, and once it binds copper it may have an anti-inflammatory effect with suppression of free radicals, formation of thromboxane, release of iron oxidant and TGFβ, increase of superoxide dismutase and vasodilation, which would be essential for tissue repair in LC. It also has a chemoattractant effect on macrophages, mast cells, and capillary cells (Da et al., 2015). All these effects are essential for the entire process of tissue repair in CL.

The wound healing process includes homeostasis, inflammation, proliferation, and remodeling. In the case of CL, once the infection is controlled by the treatment, the inflammatory phase ends and, the oxidative response and the production of proinflammatory molecules involved in tissue damage disappear. The oxidative response is regulated by a decrease of HBA1, HBD, and PRDX2. While HBA1 and HBD are essential for oxygen transportation, PRDX2 is essential for the oxidative response. The relationship between HBA1 and HBD with PRDX2 is also demonstrated in chronic inflammatory processes such as Bowel’s disease (Senhaji et al., 2017). On the other hand, FABP5 can regulate the production of prostaglandin E2 (Bogdan et al., 2018) and S100A2 is associated with inflammatory processes such as periodontitis and constitutes a candidate for biomarker of inflammation in this context (Heo et al., 2011). The repair process begins with migration, adhesion, proliferation, and cell differentiation. The GHK controls the migration of macrophages and the macrophages differentiate to M2 type to increase the production of TGFβ (Da et al., 2015). SERPINB1 attract fibroblasts to the wound healing site (Torriglia, Martin & Jaadane, 2017) and LAMP2 promotes the extracellular matrix (Li et al., 2017b). Lastly, KRT10, S100A9, and S100A8 induce proliferation and differentiation of fibroblast (Eming et al., 2010; Zhao et al., 2016) and then, occur the re-epithelialization with the participation of keratin and type II cytoskeletal 73 protein.

The identification of these proteins with increased expression after healing are closely related to the beginning of the cell proliferation stage when the infection and inflammation is controlled. Although unexpected, we do not find proteins belonging to the parasite in the ulcer center, possibly because the proportion of infected cells in the ulcer is lower than in the active borders.

The measurement by RT-qPCR showed an induction level of S100A8 > 2 in the cured patients suggesting that SERPINB1 and S100A8 are good candidates for an early biomarker of healing. In turn, the induction level of S100A9 was <2, indicating that this marker is not a good candidate. Nevertheless, before accepting or ruling out any of these candidates for biomarkers, these results need to be validated in a larger group of patients.

Conclusion

In conclusion, our work allowed for two very important analyses. The first is the characterization of proteome of pathogenesis in the ulcer generated in CL and the second is the identification of the changes in the protein profile in a positive therapeutic response to treatment, allowing to identify pathways related to inflammation and a negatively regulated oxidative response, usually associated with the reduction of inflammation and increased tissue repair. With these results, it is possible to propose a multicentric study that allows the validation of the healing biomarkers by RT-qPCR for the early prediction of the treatment. These results will allow clinicians to implement, in a timely manner, an appropriate strategy to guarantee a successful treatment.

Supplemental Information

Supplemental Information 1 Expression level.

Click here for additional data file.

Authors thank Juliana Quintero, MD, for the assistance during patients care.

Additional Information and Declarations

Competing Interests

Author Contributions

Human Ethics

Data Availability

The authors declare that they have no competing interests.

Andrés Montoya conceived and designed the experiments, performed the experiments, analyzed the data, prepared figures and/or tables, authored or reviewed drafts of the paper, approved the final draft.

Manuel Carlos López conceived and designed the experiments, analyzed the data, contributed reagents/materials/analysis tools, approved the final draft.

Ivan D. Vélez analyzed the data, contributed reagents/materials/analysis tools, approved the final draft.

Sara M. Robledo conceived and designed the experiments, analyzed the data, contributed reagents/materials/analysis tools, prepared figures and/or tables, authored or reviewed drafts of the paper, approved the final draft.

The following information was supplied relating to ethical approvals (i.e., approving body and any reference numbers):

Human Bioethics Committee of Universidad de Antioquia–SIU approved this study (Act 007-2017).

The following information was supplied regarding data availability:

Raw data are provided in a Supplemental File.

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
