# Peer review of "Label-free quantitative proteomic analysis reveals potential biomarkers for early healing in cutaneous leishmaniasis"

_PeerJ, doi:10.7717/peerj.6228_

## Round 0.1 · original submission · Major Revisions

The review process is now complete, and two thorough reviews from highly qualified referees are included at the bottom of this letter. All reviewers including myself agree the manuscript deserves to be published. Although there is considerable merit in your paper, we also identified some concerns that must be considered in your resubmission. Please, provide explanation to the analysis performed and details and justification of samples used.

Reviewer 1 ·

Basic reporting

The manuscript (#31402) submitted to PeerJ, entitled “Label, free quantitative proteomic analysis reveals potential biomarkers for early healing in cutaneous
Leishmaniasis”, whose aim was to compare the proteomic profiles in patients with Cutaneous Leishmaniasis (CL) before and after 7 days of treatment and identify early biomarkers of curative response. Twelve proteins showed differential expression in the comparative LC-electrospray ionization MS/MS (LC-ESI-MS/MS) by triplicate analysis. Seven of them were up-regulated and five of them were down-regulated. It was confirmed that three of the seven up-regulated proteins also had increased expression of their mRNA.

The purpose and approach are quite interesting in the area and the manuscript may attract the attention of several readers. However, several points must be clarified.

The authors state that “The aim of this study was to find and validate early biomarkers of therapeutic response in CL using proteomics without labeling in order to provide patients with adequate monitoring and identify early treatment failure, thus providing the possibility of changing the therapeutic scheme in a timely manner.” However, there are no validation tests of the biomarkers, only an mRNA quantification of 3 candidate proteins, whose increased expression was confirmed. Also, how could they identify early treatment failure if there are no patient samples with this characteristic?

Experimental design

Another very important point that needs to be better described is in relation to the sample. The number of patients and samples do not seem clear to me. It was also not possible to understand whether technical or only biological triplicates were made. In METHODS section, line 99: “ Four patients with a parasitological diagnosis of leishmaniasis with confirmation of species by PCR-RFLP were recruited.” Shortly thereafter, line 189 “Six patients were included in the validation assay.” In RESULTS section, line 240: “three of the four biological replicas”, and still in line 288 “All 5 patients were cured after 6…”
The experimental design is not clearly described. It would be useful to include a schematic describing the experimental design, or to describe this more clearly the text. How many patients, how many samples of each patient were prepared and loaded on the gel? And on the LC-MS/MS, or on the mRNA?

It can be seen from the title as well as in the M & M that label-free proteomic approach was performed. However, the authors seem to make some mistake in the section results quoting label-based quantification (Line 240-241, line 267).
As is the standard in proteomics work, I suggest that identified proteins be categorized according to GO. The GO defines concepts/classes used to describe gene function, and relationships between these concepts. It classifies functions along three aspects: molecular function, cellular component and biological process.

Validity of the findings

In RESULTS section, What is the correct HDB (line 258) or HBD (line 283)?
About DISCUSSION section, it could be very interesting for the authors to define a touchstone for the proteins that will be considered in this section. As it is written, it was not possible to understand why some proteins are there, for example Keratin, type II cytoskeletal 73 and LAMP2, KRT10 and ATP7A.
In addition, it may be that the lines 326-332 are dispensable.

Figures 2A as well as 3B are dispensable.
Figure 4 - it might be interesting to note if the proteins are up or down regulated.
Caption Figure 5 is so confuse. Rewrite.

Reviewer 2 ·

Basic reporting

The manuscript titled “Label-free quantitative proteomic analysis reveals potential biomarkers for early healing in cutaneous leishmaniasis” aims a critical issue regarding the disease combat. The manuscript is well written and minor correction in English are required. The data is showed correctly in the tables and figures. The authors applied cutting edge technology to search for biomarkers and reveal new proteins for future research.

Experimental design

The experimental design was well drawn; the distinct stages of the work were carefully and diligently put together. And overall the experimental design is ok! The material and methods section is well written and have sufficient details to be replicated if necessary.

Validity of the findings

In my consideration, there are some important questions not addressed by results and not discussed in the manuscript; The proteome of the leishmania parasite is been found to hold almost 50% of hypothetical/Uncharacterized proteins and it is surprisingly that in a wide proteome analysis, no one of these proteins was found. The authors cited an article (reference 5) in which hypothetical proteins were assayed. Another concern is about statistical validity of the results, the authors performed all the assays with a very low number of patients. These facts should be stated in a paragraph in the discussion section.

Additional comments

The manuscript entitled "“Label-free quantitative proteomic analysis reveals potential biomarkers for early healing in cutaneous leishmaniasis” represents a nice piece or work and a good contribution to the of new biomarkers. The manuscript is well written and organized.

---

## Round 0.2 · accepted · Accept

The authors have well addressed all the suggestions and recommendations raised by the Reviewer.

Reviewer 1 ·

Basic reporting

I believe that, following the revisions, in the current version the article is suitable for publication

Experimental design

I believe that, following the revisions, in the current version the article is suitable for publication

Validity of the findings

I believe that, following the revisions, in the current version the article is suitable for publication

Additional comments

I believe that, following the revisions, in the current version the article is suitable for publication